# Primary School Teachers’ Perspective of Sexual Education in Spain. A Qualitative Study

**DOI:** 10.3390/healthcare9030287

**Published:** 2021-03-05

**Authors:** Fernando Jesús Plaza-del-Pino, Isabelle Soliani, Cayetano Fernández-Sola, Joaquín Jesús Molina-García, María Isabel Ventura-Miranda, María Ángeles Pomares-Callejón, Olga María López-Entrambasaguas, María Dolores Ruiz-Fernández

**Affiliations:** 1Department of Nursing, Physiotherapy and Medicine, University of Almeria, 04120 Almeria, Spain; ferplaza@ual.es (F.J.P.-d.-P.); mvm737@ual.es (M.I.V.-M.); mpc630@ual.es (M.Á.P.-C.); mrf757@ual.es (M.D.R.-F.); 2Almeria Colegio Internacional, 04007 Almeria, Spain; isabellesoliani@gmail.com; 3Facultad de Ciencias de la Salud, Universidad Autónoma de Chile, Santiago 7500000, Chile; 4Facultad de Ciencias de la Educación, University of Almeria, 04120 Almeria, Spain; joaquinmolinag@gmail.com; 5Departament of Nursing, University of Jaén, 23071 Jaén, Spain; omlopez@ujaen.es

**Keywords:** sexual education, school, teachers’ perspective, qualitative research

## Abstract

Sexual education is a part of the teaching-learning process that addresses cognitive, psychological, physical and social aspects of sexuality. The purpose of sexual education is to provide people with knowledge, abilities, attitudes and values that will help them to have good sexual health, well-being and dignity. The objective of this study was to explore the perspective of primary school teachers regarding Sexual Education in school. A descriptive qualitative study was designed based on content thematic analysis. Fifteen open-ended interviews with primary school teachers were carried out, followed by inductive data analysis using ATLAS.ti software. Two key themes emerged from the analysis: “In search of a comprehensive approach to Sexual Education” and “Barriers to Sexual Education in schools: From the lack of training to fear of the families”. We conclude that despite the efforts to implement a comprehensive approach to Sexual Education that recognises sexuality as a right, primary school teachers face difficulties in delivering Sexual Education in schools due to a lack of training and the fear that parents will reject their children being spoken to about sexuality.

## 1. Introduction

The United Nations Educational, Scientific and Cultural Organization (UNESCO) defines comprehensive sexuality education as part of the teaching-learning process that addresses cognitive, psychological, physical and social aspects of sexuality [1]. The purpose of this process is to provide people with knowledge, abilities, attitudes and values that will help them to have good sexual health, well-being and dignity; establish social and sexual relations based on respect; analyse the repercussions of one’s decisions on one’s own well-being and that of others; and understand how to protect one’s rights [1]. Moreover, the World Health Organization (WHO) defines sexual health as a state of physical, emotional, mental and social well-being in regard to sexuality, not just as the absence of illness or of dysfunction or weakness [2]. This “social” connotation of the concept of sexual health requires a positive and respectful approach to sexuality that takes into account the community and family context [3,4,5] and includes sexual relations, sexual pleasure and safe sexual experiences, free from coercion, discrimination and violence [2].

Recent studies in Spain have brought to light the shortcomings of affective sexual education amongst school students, who access information about sexuality through social networks and internet pornography [6]. This can contribute to the early onset of sexual activity [7] and a distorted perception of relationships, contributing to the imitation of predominant chauvinist and violent male attitudes reflected in the pornography industry [6].

The introduction of Sexual Education (SE) in schools has been linked to the decrease in behaviours that reflect inequality [8], fewer unstable relationships [9], positive development in terms of risk prevention and the promotion of sexual health [10] and a reduced association of sexual relations with other high-risk behaviours—such as drug consumption [9]. However, there are obstacles to SE, such as parents’ reluctance to talk openly about sexuality with their children [11,12], peer pressure [5,13] and the heavy influence of the media’s representation of sexuality [11].

Despite its proven benefits [4,14,15], SE in schools remains an unresolved issue in many Western societies [16,17,18]. There is ongoing controversy regarding the ideal setting for this type of education to take place (i.e., at school, or home) [19] as well as its content, format and nature in relation to school [12,14,16]. In the school setting, the debate focuses on whether to include SE as a subject [1] or keep it as a transversal content [20]. Literature in the field reports numerous innovative programmes that tend to fail due to primary school teachers’ fear of discussing the topic openly and firmly [18,21]. Some studies suggest a lack of knowledge, which is occasionally compensated for by the assistance of healthcare professionals [22] or medicine students [23]. In terms of content, some authors cite contents linked to health concerns related to sexuality—such as unwanted pregnancies [14,24] or the excessive weighting of reproductive health in Science [21]—which results in certain topics being excluded, such as sexuality as a right, [1] sexual diversity [25] or gender issues. In an attempt to overturn these shortcomings, various innovative programmes have been developed that have a holistic vision of SE in schools that goes beyond sexual reproduction, with the aim of addressing the body, feelings, etc. [21]. Likewise, there are SE programmes based on life skills [26] or positive values that offer students the confidence and knowledge required to protect themselves and make informed decisions [15,22,23]. In Asturias, Spain, a programme developed by external experts had a temporary effect on protective behaviours, thus suggesting that one-off interventions have a limited impact [27].

In some countries and various states of the United States, SE is obligatory in public schools and is delivered by primary school teachers or nurses [28,29]. In Europe, Finland and Sweden, SE is part of the school curriculum, which has led to positive developments in preventing risk and promoting sexual health, although society and students alike have deemed it insufficient, inefficient and unadapted [10,30]. In the United Kingdom, Sexual Education in schools is not a homogenous issue; it varies from a lack of regulation in Scotland to being primarily focused on biological aspects (e.g., anatomy, infections) in England [31]. The Dutch SE programme is considered the most comprehensive and liberal in the world. It aims to foster self-sufficiency, self-respect and respect for others [32]. In Spain, teachers suggest working on coeducation as a cross-curricular content [33].

Concern surrounding pregnancy and sexually transmitted diseases continues to permeate SE programmes, which tend to define sexual health as reproductive health [11,32]. This leads to parents having the idea that SE should promote abstinence as the best option for young people [34]. Sexually transmitted infections are a serious problem for sexual health due to the way in which sexual relations are taking place [35]. However, there is increasing emphasis placed on other aspects such as coeducation or studies about stereotypes. In fact, research conducted in Spain highlights the persistence of sexist behaviours and stereotypes amongst students in elementary education [33] and sexist prejudice amongst future primary school teachers [36]. 

Although there are studies that focus on the results of teaching innovation [21] as well as how to work with parents on the topic of SE [37], there is a scarcity of literature that delves into the perspectives of primary school teachers regarding SE in schools in Spain [33]. Exploring the perspectives of primary school teachers could contribute to understanding their own attitudes, prejudice, limitations and tools for teaching SE in schools [27,36]. The objective of this study is, therefore, to explore the perspective of primary school teachers regarding SE in schools in Spain.

## 2. Materials and Methods

### 2.1. Design

This is a descriptive qualitative study based on content thematic analysis [38]. This design is based on a naturalist research paradigm that aims to understand phenomena from the point of view of their protagonists [39].

### 2.2. Participants and Setting

This study was carried out in four public schools in southern Spain. The schools had six years of primary education (1st to 6th grade of primary) with an average of two classes per year group and 60 primary school teachers (average of 15 per school). The participants were chosen using convenience sampling, with the following inclusion criteria: to be primary school teachers with teaching experience in ES, thus excluding those without experience in regulated education. Prior to the study, the participants were contacted by the researchers who carried out the interviews and were invited to participate in the study. The final sample was composed of 15 active primary school teachers. Four teachers did not want to be interviewed, claiming a lack of time on the proposed dates, thus resulting in them being excluded from the process. The average age of the interviewees was 43.4 years (standard deviation 11.85) with average teaching experience of 17.8 years (standard deviation 10.41). The sociodemographic data can be found in Table 1.

### 2.3. Data Collection

Data collection took place between June and July 2017, via open-ended in-depth interviews. Participants were recruited via an email invitation, in which the conditions and objectives of the study were explained. Subsequently, an appointment was made with the individuals who gave consent to participate in the study.

In order to ensure natural and open-ended interviews in which the participants felt they could express themselves openly, the researchers received training and practised the interview techniques before conducting the interviews.

An interview protocol was used, providing information about the objective, ethical issues, consent and questions to guide the conversation (Table 2). The average duration of the interviews was 44 min. The interviews were recorded on a digital device with prior consent from the interviewee. They were transcribed immediately after taking place so that they could be revised and enhanced with the researcher’s notes. 

### 2.4. Data Analysis

ATLAS.ti software (ATLAS.ti Scientific Software Development GmbH, Berlin, Germany) was used for data analysis, integrating the transcriptions, the codifying and categorization system and memos in a project. Thematic analysis was carried out following an inductive strategy (the themes are not based on a literary review but instead emerge from the data), in compliance with the steps described by Mayring (2000) and adapted for psychoeducational and health research [39,40]. 

First step: Selecting the object of analysis within a communication framework. The research group started by defining their theoretical, professional or scientific stance [40]. In our study we used Pender’s Model of Health Promotion (MHP) based on the importance of developing the cognitive process in order to change behaviour [41] as our theoretical lens. The MHP allows for the understanding of human conduct related to health, pointing towards healthy behaviours that are not only based on preventing sexually transmitted infections or pregnancies but also, in line with the WHO [2], address pleasure and safe sexual experiences that are free from coercion, discrimination and violence. 

Second step: Preliminary analysis. In this stage, the transcripts were read with the aim of gaining a general understanding and holistic vision of the texts.

Third step: Defining the units of analysis. Through a careful re-reading of each document, significant sentences or extracts were chosen as units of analysis. We aimed to select units of analysis sufficiently large to be considered as a whole but small enough to be a relevant meaning unit during the analysis process [42]. A total of 197 quotes were selected as units of analysis and were codified in the following phase.

Fourth step: Codification: establishing rules of analysis and classification codes. The ‘open coding’ function of ATLAS.ti software was used to assign codes to the units of analysis, giving them meaning. The “insert comment code” function was applied to define the usage rules of each code as well as the requirements for successive codification of units of analysis with each code.

Fifth step: Categorization. The codes were analysed semantically to explore their meaning and group them into themes and subthemes, using the ATLAS.ti function of “group codes” and “link codes”. In order to define the emerging themes, conceptual and explanatory memos were created. 

Sixth step: The final integration of the findings. In qualitative analysis, the wording and revision of the conclusions are analysed. In our final report, we used themes, memos and codes, allowing us to achieve robust results with the support of all previously carried out work.

### 2.5. Rigor

Reliability was achieved through researcher’s triangulation throughout the process; various researchers revised the entire coding and categorization system, discarding the themes and subthemes for which there was not general consensus. To increase the internal validity, all opinions and experiences were represented, and the transcripts and analysis were returned to some participants for them to confirm the content and the researchers’ interpretations. To improve the trustworthiness of the study, was used a checklist that includes different phases of preparation, organisation and reporting of the study [42]. 

### 2.6. Ethical Issues

The study was approved by the University of Almeria’s Ethical and Research Committee (ENFISMED15-17). All participants were informed of the objective of the study, they participated voluntarily, and they gave written consent to participate in the study. They had the possibility to end the interview at any given moment. In order to ensure anonymity and confidentiality, the participants were identified using an alphanumeric code. All data have been handled in accordance with European and Spanish data protection laws.

## 3. Results

Two themes emerged from the interview analysis that allowed us to understand primary school teachers’ perspectives regarding SE (Figure 1: Conceptual map of emerging themes). Results are presented following the structure: Name of the theme, Definition/explanation of the theme. Name of the Subtheme, explanation, representative quote. The sequence explanation–quotation is repeated throughout each subtheme until the latter was fully explained. 

### 3.1. In Search of a Comprehensive Approach to Sexual Education

For the participants of the study, SE is an important factor in improving students’ quality of life given that they are sexual beings from birth. Educating individuals on this topic from a young age can not only contribute to their full and satisfying exercise of sexuality in future adult life but also to their possession of more tolerant attitudes based on equality. Moreover, the participants consider that sexuality not only addresses genital and reproductive matters but also encompasses a holistic vision of body image, how we have relationships with our peers and our feelings during those relationships. 

#### 3.1.1. The Predominance of Preventative Sexual and Reproductive Education Aimed at Young People

Primary school teachers do not address the contents of SE with their students in a natural way. They admit that, according to Spanish legislation, it is a transversal topic similar to education in road safety or anti-discrimination, for example. The majority of the interviewees avoid terms such as coitus, masturbation, etc., referring to them in an evasive or euphemistic way. For example, two participants avoided the topic in the following way: 

*SE (blushing), well…is information about…that topic* [avoiding the term “sex”]*, right?*(E1)


*It’s a complicated issue…I’m a bit afraid of addressing it with children.*
(E11)

The interviewees believe that the ideal timing to receive affective sexual education is during adolescence. Whilst topics of basic anatomy and emotions, not explicitly related to sexuality, are addressed with very young children, SE is totally absent from the primary school stage. It is only in teenage years that the issue is dealt with again in high school through specific programmes such as the Andalusian Health Ministry’s Health Education Programme ‘Forma joven’.


*(…) At a later age, from 10 to 14, they change dramatically, they start having their first sexual encounters and I think that is when you should start talking to them about sexuality.*
(E14)

The main theme that emerged from the analysis of the participants’ comments regarding teaching sexuality to children is coitus and its negative consequences. Primary school teachers are concerned about sexually transmitted diseases, pregnancy and issues related to contraception, as confirmed by two participants:


*[I talk to them] about sexual relations, how both members must want to take part, but most of all, the importance of being careful, given that they could face problems. They shouldn’t make decisions lightheartedly but rather contemplate their actions carefully.*
(E3)


*Basically, the contents that allow us to drum into them the consequences of not using protection: disease, pregnancy etc.*
(E2)

When referring to the contents relating to sexuality, primary school teachers believe that their inclusion in the Science curriculum suffice, even though they only refer to reproduction and anatomical matters.


*I would include sexual organs, safe and consensual sexual practices, prevention of disease and pregnancy…I don’t know, things like that.*
(E4)


*It is already included in the Science curriculum, there is an entire unit dedicated to reproduction and the reproductive system.*
(E13)

#### 3.1.2. The Blueprint for Affective Sexual Education Based on Rights

Some participants understand that SE in schools should not merely address biological and reproductive contents but should instead have a more holistic vision of sexuality, including sexuality as a human right (diversity, respect, consensuality, etc.), communication (e.g., interpersonal relationships, safe sexual experiences) and the emotional aspects (e.g., pleasure).


*That’s it, I’d talk about prevention but also about how to get involved in relationships, about respect…*
(E15)


*I would focus on Sexual Identification, introduce feelings, empathy towards different sexual tendencies (…), something that would help them to learn and respect everyone, however they may be.*
(E6)

The teachers are also very concerned about gender equality, given that it is of high importance nowadays. They are of the opinion that we should dismantle stereotypes seen in the media, especially on television or in social media. In class or at break time, children tend to act in a way that reflects the clichés seen in the media. The teachers feel obliged to tackle the issue.


*…one must promote respect. They see how women are treated as objects on TV programmes and then they demonstrate these attitudes in class. For us it’s really hard to fight against what they see on TV.*
(E10)


*They copy everything they see…the most dangerous thing is that they are watching pornography on their mobiles at an ever-younger age and they think that relationships are like that.*
(E12)

### 3.2. Barriers to Sexual Education in Schools: From the Lack of Training to Fear of the Families

One of the most common limitations, according to the teachers, is the lack of training on the topic of sexuality. This is noted both at university education level, in which childhood sexual development is not addressed, as well as in a lack of training on the specific topic itself. Yet, there is training in the use of interactive whiteboards, the bilingual curriculum and similar topics. Moreover, the teachers expressed their apprehension regarding the students’ families, not only in having to speak to them clearly and decisively but also out of fear of the potential action families could take against them. This, in turn, leads to SE programmes failing.

#### 3.2.1. Lack of Training in a Comprehensive Approach to Sexuality 

The interviewees repeatedly refer to a shared complaint; the absence of training in SE throughout their university degrees, stating that university programmes are full of contents involving procedural knowledge, leaving little time to address contents more specifically related to SE. Two participants highlighted this in the following way: 


*In my degree, they didn’t even tell me that I had to address this topic with the kids in class.*
(E11)


*It’s complicated, we already have a heavy workload to have to then educate children in a topic for which we have not been prepared, to be honest (worried face).*
(E9)

Furthermore, in schools, aspects related to sexuality are included in a set of transversal contents such as healthy eating, gender equality and education for peace. Added to the lack of specific training in the field, it becomes more difficult to evaluate these contents. 


*The thing is that it’s easy to say “look, let’s put this topic into the transversal contents”, but they are contents that cannot be evaluated, unlike Maths. This means that if the matter comes up, we address it but if not, we don’t.*
(E08)


*Ultimately, transversal contents get lost in different subjects and they don’t end up being addressed.*
(E11)

#### 3.2.2. Avoiding the Topic of Sexuality out of Fear of the Families

The teachers express their apprehension of having to address the topic of SE due to the families. Added to the lack of training they receive on the subject, there is the constant scrutiny of parents and the potential outcomes their opinions can have on the teachers.


*You sometimes don’t do anything because the families are on your back observing your every move. On many occasions, a family has come and said, “who are you to say that to my child?” If that happens in subjects like English language, imagine if you start tackling issues related to sexuality.*
(E7)

Normally, if a topic related to sexuality arises in class, whether it is due to a conflict in the playground or simply from questions in class, the teachers avoid responding out of fear of what the families might say.


*I steer clear of those topics. If it comes up in class, I try to change the subject and avoid answering, just because of what they might say at home later.*
(E9)


*This leads us to leave those topics for certain school subjects, (…) we don’t have a lot of time anyway.*
(E1)

## 4. Discussion

The objective of this study was to explore the perspectives of primary school teachers regarding SE in schools in Spain. A qualitative approach to the study has allowed us to study these perspectives in detail using a wide and varied sample [36]. Using MHP [41] as the study’s lens, allows us to focus on SE in schools as a way of teaching health education that incorporates wellbeing and not just the absence or prevention of disease [2]. This facilitates a more comprehensive approach, preferred by students [12], which includes emotional and social aspects such as preventing bullying or promoting healthy relationships [15]. 

Our results suggest that SE refers to a complex transversal content in Spanish schools, that is often excluded completely or only addressed in Science. Furthermore, a recent qualitative synthesis suggests that sex is a powerful topic that can give rise to intense emotions, reactions and feelings; however, it is taught in the same way as other subjects in schools [31]. By contrast, in recent years, the United States is launching more inclusive programmes that explore sexual health in terms of its physical, social, emotional, intellectual and spiritual dimensions [23,24]. This study suggests that, despite the obstacles that primary school teachers face, they try to deliver SE based on rights. For this reason, using external experts (for example, sexologists), could be of great help [27]. 

In accordance with current literature on the topic, our study highlights that the common approach to this type of education is based on abstinence, prevention and reproduction [14,21], and that it is awkward, unhelpful and often uses scare tactics [43]. Various authors have concluded that SE based on abstinence does not reduce sexual activity itself or when young people start to engage in it [12,44]. Moreover, society’s view of sexuality differentiates gender roles. For example, virginity is seen as a virtue in women whereas it is considered a flaw in a man [45]. However, it is possible to modify traditional stances due to socio-cultural changes related to gender identity and ideology [46]. A comprehensive approach to SE does not only prevent disease and unwanted pregnancies but also prevents homophobic bullying, domestic violence and child abuse [15]. Furthermore, it promotes healthy relationships, socioemotional learning and media literacy [15,47]. The participants of the study argue that the lack of training and the fear of the families are barriers to a comprehensive and open-minded approach to sexuality in schools. 

Our study was influenced by the values and beliefs prevalent in Spanish society. Within this context, there has also been a change in values related to sexuality, concurring with other studies that allude to socio-political changes with important repercussions on SE [48]. Likewise, in the United Kingdom the topic has evolved from immoral to normal over a 70-year period [23,49]. 

On the other hand, our results reflected a concern amongst primary school teachers regarding gender equality, as they find themselves in the difficult position of breaking down stereotypes learned from the media, social networks and early age pornography consumption, as stated by Ballester and Orte [6].

Our study reflects that primary school teachers often lack the determination required to speak about sexuality openly, which is in line with other studies that affirm that SE can fail due solely to the fear of speaking frankly [21]. According to young people in several studies, schools struggle to accept that some young people are sexually active, leading to a lack of relevant content for them in SE [31]. The information era allows children to be exposed to various sources that reflect nudity, identity and sexual conduct continuously, underlining the importance of addressing these issues with adapted language but in an open way [50]. Moreover, it is necessary to include content regarding the possibilities, limitations and risks of social media in relation to sexual activity and behaviours [47].

A vital datum from this study is the lack of SE training amongst teachers, leading to shortcomings when addressing this subject in schools. The teachers are aware of this deficit and stress the importance of specific training in sexuality, which is a stance supported by various authors who also value the need for teachers to receive specific SE training [51,52,53] in programmes directed at all levels, including pre-adolescence [54]. The latter is perhaps the most important age to educate children on sexuality, with the support of their families.

One of the key barriers to SE alluded to in this study is the tendency to avoid talking about sexuality out of fear of families with opposing views to teachers speaking to their children about sexuality. In this respect, other experiences suggest the need to take into account the SE received by parents [11], religious beliefs [5,55] and the possibility of resistance against education in this area [56]. 

A further cause of concern regarding SE are children with specific education needs (SEN). Literature suggests that parents with autistic children are not aware of whether their children have experienced orgasms [57]. There is a notable increase in quality of life amongst these children after receiving SE, as it enables them to understand differences in identity and sexual practices [58]. However, this concern did not arise in our study, perhaps due to the context in which the study took place or to the participants’ lack of SEN students.

Both in this research study and others [35] it is claimed that increasing knowledge of sexual health and sexuality leads to a more aware approach towards the understanding and risks of sexual encounters. Literature supports the notion that affective sexual education teaches children to protect themselves from mistreatment, in so much as they are aware of appropriate and inappropriate behaviours. They also advocate that SE gives children the knowledge required to increase protective behaviours against abuse [59]. Likewise, incorrect and aversive beliefs about SE can be changed [35] not only in schools but throughout life, given that childhood and adolescence are not the only stages in life that one should receive formal SE as it should continue into adulthood [29]. This study suggests including aspects related to sexuality across all subjects and not only in Science. This would contribute to a holistic approach to SE based on rights and not just on prevention. For example, subjects like Spanish Language could explore non-sexist language or History could look into the historical causes for gender discrimination. 

Some of the limitations of this study are related to the preparation phase, specifically the sampling section [42]. The sample is appropriate and was composed of participants with knowledge of the research topic. However, the sample was selected in public schools in a middle-class urban setting. This leads us to believe that extending this study to primary school teachers in private or semi-private schools as well as other more varied settings could lead to a more robust set of results. It is a heterogeneous sample in terms of age (range: 25–60 years) and professional experience (1–40 years). This has provided us with a diversity of perspectives, but a more homogenous sample would have allowed us to have a deeper understanding of the difficulties associated with each individual’s experience or the generational difference when talking about sex. New studies could compare perspectives and experiences of young and older teachers in terms of speaking about sexuality with their pupils. The primary school teachers have identified their perspectives regarding SE in schools and the key obstacles that they face. Research is therefore needed into how to make them more competent.

## 5. Conclusions

Primary school teachers conclude that SE in schools has an excessively preventative approach. Nonetheless, they recognise an attempt to implement a more comprehensive SE that includes affective sexual education and sexuality as a human right.

The teachers are faced with difficulties when addressing SE in schools due to a lack of specific training in the area and fear of parents rejecting their children being spoken to about sexuality. The primary school teachers have highlighted the key barriers that they face. It is necessary to continue researching their strengths, capabilities and how to make them more competent in delivering SE in schools. Understanding formative experiences in other countries and surroundings could also be useful.

It is essential to develop specific SE training programmes for teachers and to establish educational programmes in Spanish schools that support an integrated, holistic approach to sexual health based on equality. Furthermore, it is vital to create awareness campaigns regarding the need for SE in schools given that it is a fundamental element in developing individuals who are able to make free and healthy choices concerning their relationships. These campaigns should be directed at the general population to break the taboos regarding SE for children. This could be done within institutions (e.g., governments) or from the schools themselves through parent meetings. 

Practical implications: as SE is a cross-curricular competence in the Spanish education system, it is vital that teachers address it from different perspectives. Leaving it solely to Science teachers risks SE becoming too focused on prevention. The participants stated a lack of training in the subject, which is why the support of specialists from the sexology field, for example, would be of great benefit in the design of strategies and materials for SE in schools. 

## Figures and Tables

**Figure 1 healthcare-09-00287-f001:**
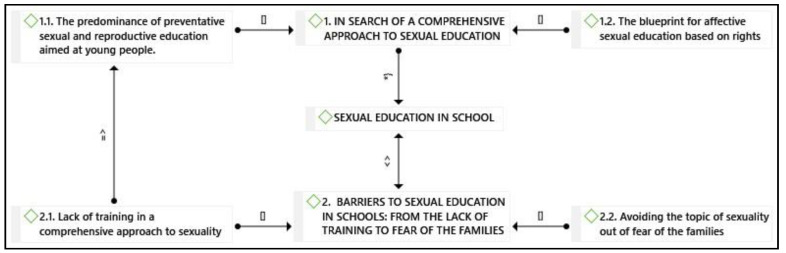
Conceptual map of emerging themes. Key: [] is part of; => is cause of; <> opposed to; *} is property of.

**Table 1 healthcare-09-00287-t001:** Sociodemographic data of participants.

Interview Code	Age	Gender	Years of Experience
E1	45	Woman	18
E2	60	Woman	34
E3	38	Man	10
E4	40	Woman	19
E5	35	Woman	11
E6	30	Man	9
E7	29	Man	4
E8	43	Woman	16
E9	51	Woman	26
E10	33	Woman	21
E11	48	Woman	17
E12	62	Man	40
E13	25	Woman	1
E14	55	Man	20
E15	57	Man	22

**Table 2 healthcare-09-00287-t002:** Interview protocol.

Stage of the Interview	Subject	Content and Example of Questions
Introduction	Our intention	I am a member of a research group about Sexual education in schools. Knowing your perspectives could help and be useful to propose improvements in SE in Spanish schools.
	Information and ethical issues	We need to record the conversation in order for the research team to analyze the data. Only the research team will have access to the recordings. Participation is totally voluntary and you can leave the study at any time you wish. Your identity will be protected, and your name and personal data will not be revealed.
	Consent	Verbal acceptance of the participants and signing of the corresponding document.
Beginning	Introductory question	As a teacher, tell me about the importance of teaching students about sexuality to you.
Development	Conversation guide	Tell me about the contents that you think sexual education includes. Which content should be included in your opinion? How do you implement sexual education? Tell me about the advantages and disadvantages or obstacles of a subject about sexual education in schools, and as cross-curricular content.
Closing	Final question	Is there anything else you would like to tell me?
	Appreciation	Thank you for taking part. Your testimony will be used for the research study. We remain at your disposal if you need anything. You will receive the study upon completion.

## Data Availability

Data of this study are audio recordings and their transcripts (confidentially), which are stored in an ATLAS.ti software Project.

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
