# Peer review of "Primary School Teachers’ Perspective of Sexual Education in Spain. A Qualitative Study"

_healthcare, 2021, doi:10.3390/healthcare9030287_

Round 1
Reviewer 1 Report
Thank you for this opportunity to review a manuscript titled “Primary school teachers´ experiences of sexual education in Spain: a qualitative study.” This manuscript has many strong points and it is well written. Nevertheless, I would like to ask the authors to address some points in order to improve the quality of this manuscript.
Abstract: The authors have defined following keywords in abstract: sexual education, affective education, health education, school, qualitative research. However, keywords “affective education” and “health education” are not used in abstract nor they are not included in the title of this manuscript. Hence, I would suggest that the authors would reconsider using these keywords. In addition, I would like to point out that there is no keyword regarding teachers which are the participants of this study.
Introduction: The topic involves innovative perspectives and it has value for development of sexual education in schools. However, I think that the authors could strengthen the introduction by describing in more detail why this scientific knowledge regarding teachers´ perspectives was needed. Now introduction focuses mainly on sexual education (SE) in schools and SE programmes. The literature available about the SE has been explored well and references are relevant, but as a reader I would like to know more why teachers´ perspectives had to be explored.
Objective of the study: The objective of this study was to explore the opinions and experiences of primary school teachers regarding SE in schools in Spain. This objective is relevant, and it is in line with the methodology used in this study. However, I would like to ask what is the relationship between opinions and experiences? Opinions are not included in the title of this manuscript. In addition, when describing results authors use the term “perspective”. Hence, I think that the objective of this study should be clarified. I would suggest that the authors would use the term “perspective” instead of opinions and experiences.
Materials and methods: The study has been planned and implemented applying research methods and selection of those methods is justified. However, there is some inconsistencies that should be addressed. For instance, in abstract it is mentioned that 12 open-ended interviews were carried out but on page 3 authors describe that final sample consisted of 15 teachers. In addition, I would ask the authors to describe in more detail what was the unit of analysis in this study as it is an essential part of a qualitative content analysis and therefore affects trustworthiness of the content analysis. Moreover, authors use the term “category” in methods but in results the term “theme” is used instead – what is the rationale behind this?
And lastly, the authors argue that reliability was achieved through data triangulation amongst the researchers throughout the process. However, I do not agree with the authors on that matter. Data triangulation means that multiple data sources are being used in a single study. Hence, I would say that investigator or researcher triangulation was used in this study.
Limitations: I would like to ask the authors to critically review the limitations of this study. Now limitations are reviewed quite superficially on page 8. In addition, the authors could use references to support their argumentation regarding the trustworthiness of this qualitative study. For instance, it would be good if the authors used checklists developed for these purposes (aim to improve the trustworthiness of a content analysis). Here is one tip for the authors: Elo S, Kääriäinen M, Kanste O, Pölkki T, Utriainen K, Kyngäs H. Qualitative Content Analysis: A Focus on Trustworthiness. SAGE Open. January 2014. doi:10.1177/2158244014522633.
Conclusions: The conclusions are justified based on the results. However, it would be a great value for this manuscript if the authors could provide some recommendations for developing teachers´ competencies in this area, as well as how to support teachers in sexual education. Now conclusions mainly focus on developing SE in schools. Teachers´ perspective should be argued in more detail since it is the core of this study.
To sum up, the structure of the manuscript is consistent. Clarity and readability are good.
Author Response
Response to Reviewer 1.
Thank you for this opportunity to review a manuscript titled “Primary school teachers´ experiences of sexual education in Spain: a qualitative study.” This manuscript has many strong points and it is well written. Nevertheless, I would like to ask the authors to address some points in order to improve the quality of this manuscript.
Abstract: The authors have defined following keywords in abstract: sexual education, affective education, health education, school, qualitative research. However, keywords “affective education” and “health education” are not used in abstract nor they are not included in the title of this manuscript. Hence, I would suggest that the authors would reconsider using these keywords. In addition, I would like to point out that there is no keyword regarding teachers which are the participants of this study.
Response: The keywords “affective education” and “health education” have been deleted. The keyword “teachers” has been added to improve the paper’s consistency. Thank you for the suggestion.
Introduction: The topic involves innovative perspectives and it has value for development of sexual education in schools. However, I think that the authors could strengthen the introduction by describing in more detail why this scientific knowledge regarding teachers´ perspectives was needed. Now introduction focuses mainly on sexual education (SE) in schools and SE programmes. The literature available about the SE has been explored well and references are relevant, but as a reader I would like to know more why teachers´ perspectives had to be explored.
Response: There is now more detail as to why the scientific knowledge regarding teachers’ perspectives was needed. Examples of prior research in Spain have been included about the need to explore teachers’ perspectives (p.1 and 2)
Objective of the study: The objective of this study was to explore the opinions and experiences of primary school teachers regarding SE in schools in Spain. This objective is relevant, and it is in line with the methodology used in this study. However, I would like to ask what is the relationship between opinions and experiences? Opinions are not included in the title of this manuscript. In addition, when describing results authors use the term “perspective”. Hence, I think that the objective of this study should be clarified. I would suggest that the authors would use the term “perspective” instead of opinions and experiences.
Response: Thank you for the suggestion. We have used the term “perspective” in the title, the keywords and the objectives. This improves the study’s consistency.
Materials and methods: The study has been planned and implemented applying research methods and selection of those methods is justified. However, there is some inconsistencies that should be addressed. For instance, in abstract it is mentioned that 12 open-ended interviews were carried out but on page 3 authors describe that final sample consisted of 15 teachers. In addition, I would ask the authors to describe in more detail what was the unit of analysis in this study as it is an essential part of a qualitative content analysis and therefore affects trustworthiness of the content analysis. Moreover, authors use the term “category” in methods but in results the term “theme” is used instead – what is the rationale behind this?
Response: Thank you for highlighting this error. The abstract has been corrected, indicating that 15 interviews took place, as described in the method section and the table of sociodemographic data. The term “category” was corrected which had been used as synonym to “theme”. To make the language more coherent, we unified the term “theme” and used it throughout the paper. In the methods we have left the denomination “Fifth step: Categorization”, from which themes and subthemes emerge. In the Third step, the units of analysis are defined better.
And lastly, the authors argue that reliability was achieved through data triangulation amongst the researchers throughout the process. However, I do not agree with the authors on that matter. Data triangulation means that multiple data sources are being used in a single study. Hence, I would say that investigator or researcher triangulation was used in this study.
Response: The reviewer is right. The sentences indicating researcher triangulation has been corrected (not data triangulation) (p.5)
Limitations: I would like to ask the authors to critically review the limitations of this study. Now limitations are reviewed quite superficially on page 8. In addition, the authors could use references to support their argumentation regarding the trustworthiness of this qualitative study. For instance, it would be good if the authors used checklists developed for these purposes (aim to improve the trustworthiness of a content analysis). Here is one tip for the authors: Elo S, Kääriäinen M, Kanste O, Pölkki T, Utriainen K, Kyngäs H. Qualitative Content Analysis: A Focus on Trustworthiness. SAGE Open. January 2014. doi:10.1177/2158244014522633.
Response: The limitations of the study have been critically reviewed. We have used a checklist, as suggested, for the trustworthiness of the study (in the section rigor - p.5- and limitations – p.12). Thank you for this.
Conclusions: The conclusions are justified based on the results. However, it would be a great value for this manuscript if the authors could provide some recommendations for developing teachers´ competencies in this area, as well as how to support teachers in sexual education. Now conclusions mainly focus on developing SE in schools. Teachers´ perspective should be argued in more detail since it is the core of this study.
Response: The conclusions of the study have been revised, providing recommendations for developing teacher’s competencies and emphasising the perspectives of the teachers who took part in the study.
To sum up, the structure of the manuscript is consistent. Clarity and readability are good.
En resumen, la estructura del manuscrito es consistente. La claridad y la legibilidad son buenas.
Response: Thank you for your comment.
Reviewer 2 Report
Dear colleague/s,
First of all, I want to congratulate you on the topic of your paper: it is a very interesting and necessary work. Below I make some comments: my purpose is to make some suggestions for improvement.
- Introduction. This section includes evidence and theoretical foundations that help readers understand the topic, but also the gaps that emphasize the importance of the current study. However, most of this section is not based on current studies conducted in Spain. From my point of view, this is a major limitation of the work: the justification of the objectives could be weak. Consequently, to support the justification of the objectives of this paper, I suggest an in-depth review of the research carried out in Spain.
- Materials and Methods. I consider that this section needs much more detail. For example, how was the interview designed and validated? Were the themes and the subthemes based on the theoretical review? Were all they emerging? In addition, it is perhaps not clear how informed consent were obtained from the participants (i.e. what information the participants had about the characteristics and objectives of the study).
- Results. The analyzes performed are adequate and results are presented clearly.
- Discussion. This paper only provides surface discussion, analysis and exploration of the results: from my point of view, stronger arguments are needed. In addition, it is necessary to point out the limitations of the work as well as future lines of work based on the results of this study.
- Conclusions. The practical implications of the study are weak and remains at a perfunctory level.
- Appendix. It might be useful to detail the questions that make up the in-depth interview.
Again, I appreciate the effort to explore this area of research. However, I would encourage the author(s) to step back, and consider how to make this paper more compelling and serve as a greater contribution to the field of research. As previously said, I hope these comments help improve an interesting paper.
Kind regards,
Author Response
Response to Reviewer 2.
First of all, I want to congratulate you on the topic of your paper: it is a very interesting and necessary work. Below I make some comments: my purpose is to make some suggestions for improvement.
Introduction. This section includes evidence and theoretical foundations that help readers understand the topic, but also the gaps that emphasize the importance of the current study. However, most of this section is not based on current studies conducted in Spain. From my point of view, this is a major limitation of the work: the justification of the objectives could be weak. Consequently, to support the justification of the objectives of this paper, I suggest an in-depth review of the research carried out in Spain.
Response: Thank you for your comment. A more in-depth review of studies conducted in Spain has been carried out. We have included some new studies that highlight previous work on this topic in Spain.
Ballester, Ll.; Orte, C. Nueva pornografía y cambios en las relaciones interpersonales, 1fst ed.; Octaedro: Barcelona, Spain, 2019.
García-Vázquez, J.; Quintó, L.; Agulló-Tomás, E. Impact of a sex education programme in terms of knowledge, attitudes and sexual behaviour among adolescents in Asturias (Spain). Glob Health Promot 2020, 27, 122-130. DOI: 10.1177/1757975919873621.
González, E.; Rodríguez, Y. Estereotipos de género en la infancia. Pedagogía Social. Revista Interuniversitaria 2020, 36, 125-138. DOI: 10.7179/PSRI_2020.36.08.
Heras-Sevilla, D.; Ortega-Sánchez, D. Evaluation of sexist and prejudiced attitudes toward homosexuality in Spanish future teachers: analysis of related variables. Front Psychol 2020, 11, 2334. DOI: 10.3389/fpsyg.2020.572553.
Materials and Methods. I consider that this section needs much more detail. For example, how was the interview designed and validated? Were the themes and the subthemes based on the theoretical review? Were all they emerging? In addition, it is perhaps not clear how informed consent were obtained from the participants (i.e. what information the participants had about the characteristics and objectives of the study).
Response: More detail is provided in the section “Materials and Methods”: Information about participant recruitment and about the context of the study. The interview protocol is included in Table 2 in which one can see the information provided to the participants as well as the questions that served as a guide in the interviews (p. 3).
We indicate that we followed an inductive analysis strategy in which themes emerge from the data, which are not based on the literature review (p. 4)
Results. The analyzes performed are adequate and results are presented clearly.
Response: Thank you for your comment.
Discussion. This paper only provides surface discussion, analysis and exploration of the results: from my point of view, stronger arguments are needed. In addition, it is necessary to point out the limitations of the work as well as future lines of work based on the results of this study.
Response: The “Discussion” section has been re-written, indicating the importance of research with specific reference to our results. The limitations of the study have been highlighted along with proposals for future research (p. 10).
Conclusions. The practical implications of the study are weak and remains at a perfunctory level.
Response: We have improved the practice implications of this study. (p. 10)
Appendix. It might be useful to detail the questions that make up the in-depth interview.
Response: The interview protocol has been included (Table 2) in which you can see the information provided by the participants and the questions that were used to guide the interviews.
Again, I appreciate the effort to explore this area of research. However, I would encourage the author(s) to step back, and consider how to make this paper more compelling and serve as a greater contribution to the field of research. As previously said, I hope these comments help improve an interesting paper.
Response: Thanks again to the reviewer for the suggestions and comments. The manuscript has been revised taking into account the suggestions. We hope to have improved it considerably.
Table 2. Interview protocol
|
Stage of the interview |
Subject |
Content and example of questions |
|
Introduction |
Our intention |
I am a member of a research group about Sexual education in schools. Knowing your perspectives could help and be useful to propose improvements in SE in Spanish schools. |
|
Information and ethical issues |
We need to record the conversation in order for the research team to analyze the data. Only the research team will have access to the recordings. Participation is totally voluntary and you can leave the study at any time you wish. Your identity will be protected, and your name and personal data will not be revealed. |
|
|
Consent |
Verbal acceptance of the participants and signing of the corresponding document. |
|
|
Beginning |
Introductory question |
As a teacher, what has been your experience with SE? Tell me about the importance of teaching students about sexuality to you. |
|
Development |
Conversation guide |
Tell me about the contents that you think sexual education includes. Which content should be included in your opinion? How do you implement sexual education? Tell me about the advantages and disadvantages or obstacles of a subject about sexual education in schools Tell me about the advantages and disadvantages or obstacles of sexual education as cross-curricular content |
|
Closing |
Final question |
Is there anything else you would like to tell me? |
|
|
Appreciation |
Thank you for taking part. Your testimony will be used for the research study. We remain at your disposal if you need anything. You will receive the study upon completion. |
Reviewer 3 Report
In reading this article it is clear that the authors are not native English speakers, but this aspect of the article is not bad. However, there were some phrases that were oddly worded, for example on page 5 "does not only address" (this should be "not only addresses") that need revising. I would recommend that the authors seek the services of a native English speaker to proofread. There were also some adjacent sentences where the same sentence structure or choice of words was used. This too may be related to knowledge of English and some help could fix this.
The more substantive problem is the quality of the write-up of the analysis, and the discussion that follows this. In relation to this there were three areas where I identified issues:
- The data analysis is described in one place as 'content analysis' and in another as 'discourse qualitative analysis'. Which is it? If the latter, what is meant by this? I've never seen an analysis described as this before - I wondered if the authors meant discourse analysis; and if so, what type? From reading the description of how the analysis was carried out, it sounds like either a content or thematic analysis has been undertaken; and this would be consistent with use of ATLAS.ti whereas a discursive approach would not. The methods section therefore needs some revision. It would also be useful to cite a source for the specific version of content/thematic analysis used.
- The Results section is seriously flawed. I am presuming that this is the write-up rather than a problematic analysis. The quoted extracts appear relevant, but the analytical commentary around these does not related well to the content within the quotes. These things should work together and the commentary following each set of quotes does not unpack the meaning embedded in the quotes.
- The discussion section does not relate well to the analysis. Conclusions are drawn without reference to how these relate to content in the analysis. Essentially, the content here is just more literature review and therefore doesn't indicate the importance of the research with specific reference to the analytical findings.
Author Response
Response to Reviewer 3.
In reading this article it is clear that the authors are not native English speakers, but this aspect of the article is not bad. However, there were some phrases that were oddly worded, for example on page 5 "does not only address" (this should be "not only addresses") that need revising. I would recommend that the authors seek the services of a native English speaker to proofread. There were also some adjacent sentences where the same sentence structure or choice of words was used. This too may be related to knowledge of English and some help could fix this.
Response: Thank you for your comment. We have corrected the examples of same sentence structure or choice of words.
Although the surnames of the authors are Spanish or Italian, one of the authors (Isabelle Soliani) is native British and taught in British schools for 5 years and has been teaching English in Spain for 4 years. Nevertheless, as the original manuscript was written in Spanish, as the reviewer has pointed out, the poor structure of some sentences were still notable despite having been translated by a native speaker.
The more substantive problem is the quality of the write-up of the analysis, and the discussion that follows this. In relation to this there were three areas where I identified issues:
1. The data analysis is described in one place as 'content analysis' and in another as 'discourse qualitative analysis'. Which is it? If the latter, what is meant by this? I've never seen an analysis described as this before - I wondered if the authors meant discourse analysis; and if so, what type? From reading the description of how the analysis was carried out, it sounds like either a content or thematic analysis has been undertaken; and this would be consistent with use of ATLAS.ti whereas a discursive approach would not. The methods section therefore needs some revision. It would also be useful to cite a source for the specific version of content/thematic analysis used.
Response: Thank you for your comments. There was indeed inconsistency in the wording of the analysis, which has now been corrected. As the reviewer suggests, it is thematic analysis. The reference for the specific version of thematic analysis used is now referenced:
[39] Fernández-Sola, C.; Granero-Molina J.; Hernández-Padilla JM. Comprender para cuidar: Avances en investigación cualitativa en Ciencias de la Salud. Editorial Universidad Almería: Almería, 2020.
[40] Cáceres, P. Análisis cualitativo de contenido: Una alternativa metodológica alcanzable. Psicoperspectivas. Individuo y Soc 2003, 2, 53-82. DOI:10.5027.
2. The Results section is seriously flawed. I am presuming that this is the write-up rather than a problematic analysis. The quoted extracts appear relevant, but the analytical commentary around these does not related well to the content within the quotes. These things should work together and the commentary following each set of quotes does not unpack the meaning embedded in the quotes.
Response: Apologies for the confusion. We indicate in the results that the quoted extract do not precede but instead come after the explanations/interpretations of the researchers. Read in this way (first the comment and then the related quote) the results become more coherent. However, the results have been reviewed so that the comments and the quotes work well together.. E.g.:
3.1.1. Subtheme 1.1. The predominance of preventative sexual and reproductive education aimed at young people.
Primary school teachers do not address the contents of SE with their students in a natural way. They admit that, according to Spanish legislation, it is a transversal topic similar to education in road safety or anti-discrimination, for example. The majority of the interviewees avoid terms such as coitus, masturbation…, referring to them in an evasive or euphemistic way. For example, two participants avoided the topic in the following way:
SE (blushing), well…is information about…that topic [avoiding the term “sex”], right? (E1)
It’s a complicated issue…I’m a bit afraid of addressing it with children. (E11)
3.The discussion section does not relate well to the analysis. Conclusions are drawn without reference to how these relate to content in the analysis. Essentially, the content here is just more literature review and therefore doesn't indicate the importance of the research with specific reference to the analytical findings.
Response: Thank you for your comment. The “Discussion” section has been re-written highlighting the importance of the research with specific reference to our results (p. 8-9).

Reviewer 4 Report
First of all I would like to thank the editors for the opportunity to review this manuscript entitled "Primary school teachers’ experiences of Sexual Education in Spain. A qualitative study ". It is an interesting manuscript for the scientific community. It presents certain that are detailed below:
1. There are certain acronyms that are not developed the first time it is written, for example UNESCO.
2. In section 2.2. it should be further explained how the participants were chosen. How many teachers were there in the school?
3. In section 2.2. It is mentioned that 4 individuals were excluded in the process, why? Didn't meet the criteria?
4. In the conclusions section, the conclusions of the study are clearly and explicitly shown, as well as future lines on the subject. In relation to one of these lines, the authors mention that it is vital to create awareness campaigns about the need for SE in schools. How could these awareness campaigns be created? Who would they be addressed to?
Author Response
Response to Reviewer 4
Thank you for the reviewer’s constructive comments that have allowed us to learn, be more careful and improve our manuscript. All of the comments have been taken into account, they have been marked in red in the new version of the manuscript and in the responses of this letter.
First of all I would like to thank the editors for the opportunity to review this manuscript entitled "Primary school teachers’ experiences of Sexual Education in Spain. A qualitative study ". It is an interesting manuscript for the scientific community. It presents certain that are detailed below:
- There are certain acronyms that are not developed the first time it is written, for example UNESCO.
Response: The manuscript has been revised and the meaning of the acronyms are given the first time they are used.
In section 2.2. it should be further explained how the participants were chosen. How many teachers were there in the school?
Response: More details will be provided about how the participants were contacted and selected. We mention that the four schools where data were collected have an average of 15 teachers. (p.3, )
This study was carried out in four public schools in southern Spain. The schools had six years of primary education (1st to 6th of primary) with an average of two classes per year group and 60 primary school teachers (average of 15 per school).
In section 2.2. It is mentioned that 4 individuals were excluded in the process, why? Didn't meet the criteria?
Response: We indicate that the “Four teachers did not want to be interviewed, claiming a lack of time on the proposed dates…” (p.3)
In the conclusions section, the conclusions of the study are clearly and explicitly shown, as well as future lines on the subject. In relation to one of these lines, the authors mention that it is vital to create awareness campaigns about the need for SE in schools. How could these awareness campaigns be created? Who would they be addressed to?
Response: In the section “Conclusions”, we highlight that these campaigns should be directed at the general population to break the taboos surrounding SE amongst children. It could be done in institutions (e.g the government) or in the schools themselves, through parent meetings.
Round 2
Reviewer 2 Report
Dear authors, From my point of view, the revised papers could be accepted in present form.Congratulations on the work done!
Kind regards